# OpenReview forum: "Learning a Dense Reasoning Reward Model from Expert Demonstration via Inverse Reinforcement Learning"
_ICLR.cc/2026/Conference — Submitted to ICLR 2026_

### Official Review · Reviewer_RjAM · 2025-11-01

**Soundness:** 2
**Presentation:** 2
**Contribution:** 2
**Rating:** 2
**Confidence:** 3

**Summary:**

This paper reframes and applies adversarial inverse reinforcement learning to the reasoning tasks of large language models. It learns a token-level dense reasoning reward model from expert demonstrations, instead of imitating styles through supervised fine-tuning. This reward model serves two complementary roles: it provides step-level feedback to optimize the reasoning policy during training, and acts as a critic to rerank sampled traces under a fixed computational budget during inference. Moreover, it prioritizes correctness over surface form, can correlate with the validity of the final answer, and locate errors within the traces. Experiments on the GSM8K dataset using Llama3 and Qwen2.5 as backbone models show that the dense reasoning reward can be used as a learning signal to elicit reasoning capabilities, and reward-guided reranking can improve predictive performance (especially for Llama-series models). Meanwhile, this work unifies training signals, inference selection, and token-level diagnostics

**Strengths:**

The token-level dense reward model learned based on adversarial inverse reinforcement learning (IRL) in this paper breaks through the limitation of separating "training signals" and "inference auxiliary tools" in traditional methods. It can not only provide step-level training feedback for LLM reasoning policies (solving the "black-box training" problem that only relies on outcome correctness) but also act as a reranker during the inference stage to screen high-quality traces under a fixed computing budget (without the need for additional training of a dedicated reranking model)

**Weaknesses:**

1. The paper adopts the "discriminator-policy" adversarial optimization framework (maxφminθ E [expert reward - policy reward]), but fails to address the inherent **signal non-stationarity** of adversarial training. After the discriminator parameter φ is updated, the reward objective for policy optimization changes immediately, which may lead to oscillations in policy training (e.g., "evaluation reward fluctuations" in the training curve). The paper only describes this phenomenon but does not analyze its causes
2. It does not test performance on other reasoning datasets (such as MATH, AIME, and other reasoning tasks like code reasoning), making it impossible to verify whether the reward model is only suitable for arithmetic reasoning or can be transferred to other harder reasoning tasks
3. It does not test the model’s performance when "the proportion of expert demonstrations is reduced" (e.g., only 50% or 30% of expert traces are provided, and the rest are replaced with noisy traces) or "expert demonstrations contain errors". This makes it impossible to prove the effectiveness of the method in "low-quality demonstration" scenarios, which are actually more common in practical applications

**Questions:**

1. Why is GSM8K chosen as the only verification setting?
2. If expert demonstrations contain 10%-20% of incorrect steps (e.g., intermediate calculation errors), will the reward model in the paper learn the wrong signals? To what extent will the model’s performance decline in this case?
3. What are the specific impacts of "signal non-stationarity" (reward objective changes caused by discriminator updates) on policy training in adversarial training? Is there any theoretical analysis or experimental evidence to prove the convergence of this adversarial framework? If convergence exists, what are the convergence conditions?
4. After removing the "adversarial framework" (training the reward model only with expert traces), "token-level reward" (replacing it with step-level reward), and "reward normalization" respectively, how will the model’s pass@k and error localization ability change?
5. Does the method face an on-policy issue—namely, can a reward model trained on one reasoning model generalize to other reasoning models?

---

> ### Author Response · Authors · 2025-11-26
>
> We thank the reviewer for their constructive feedback. We are glad to see that the reviewer agrees that the dense token-level reward model can be helpful for training, inference, and the interpretability of Chain-of-Thoughts. Additionally, we wish to extend our apologies for replying so late; running the follow-up experiments took longer than expected. **We have uploaded a new manuscript, and all our changes are highlighted in blue.**
>
> We address the reviewer’s concerns and questions below:
>
> **W1 and Q3) Non-Stationarity of AIRL**
>
> We thank the reviewer for this point. We agree that non-stationarity is an inherent characteristic of this adversarial framework and the likely cause of the "evaluation reward fluctuations" we observed.
>
> We respectfully clarify that this is a known and accepted trade-off for the sake of tractability. This approach avoids computationally prohibitive nested optimisation loops, which are unsuitable for LLMs. We acknowledge this as a limitation, stating that adversarial training yields non-stationary signals.
>
> Our primary contribution is not a new IRL algorithm, but the **adaptation** of existing adversarial frameworks (like AIRL/GAIL) to learn a **dense, token-level reasoning reward** for LLMs. The paper's core is demonstrating the *utility* of this learned reward via our four desiderata (training signal, inference-time assistance, correctness over style, and error localisation.
>
> While a deep theoretical analysis of non-stationarity is valuable, it is beyond the scope of our empirical work. Our findings show that, despite fluctuations, the framework successfully learns a functional and interpretable reward. For foundational analysis of convergence, we would direct the reviewer to the original adversarial IRL papers we built upon (Fu et al, 2019).
>
> **W2 and Q1) Only validated on GSMK8**
>
> Thank you for providing this feedback. We agree that a baseline is not enough. To this end, we have run additional experiments on a medical example, namely the **MedReason Dataset** [1], to demonstrate that our method also works in more difficult, high-stakes environments, such as medical decision-making.
>
> We ran all experiments on the original model combinations in the MedReason dataset and report the results **Appendix C (pages 23-30)** (Don't worry, it's mainly images). We can see that our results are consistent with those reported for GSM8K. While the training is noisier than with a simple dataset like GSM8K, we can see that the large models still show a strong correlation with medical correctness. Moreover, the reranking strategy still provides significant improvements, especially on pass1|N.  We find the results particularly interesting in a high-stakes environment, where verifiable rewards are not always guaranteed, and interpretability is crucial.
>
> We hope that these additional results demonstrate further the generalisability of our method.
>
> [1] https://arxiv.org/abs/2504.00993: MedReason: Eliciting Factual Medical Reasoning Steps in LLMs via Knowledge Graphs

---

> ### Author Response · Authors · 2025-11-26
>
> **W3 and Q2) Reduced and Erroneous Expert Demonstrations**
>
> That’s an interesting observation from the reviewer. We agree that in real-world applications, the expert demonstrations might be erroneous. We run additional ablation experiments on GSM8K, where some expert demonstrations are incorrect, providing evidence of a drop in performance across almost all cases. **We updated the manuscript (Appendix D.7, page 38)**
>
> **Q4) Ablation Study**
>
> We follow the reviewer’s instructions and provide an ablation study for AIRL with only an outcome reward signal and without reward normalisation in **Appendix D.5, page 36** and **Appendix D.6, page 37**, respectively. It can be seen that using only an outcome signal leads to slightly better performance. However, this comes at the cost of not having a dense reasoning reward model. In the second experiment, we observe that omitting reward normalisation results in significant instability during training.
>
> **Q5) Does the method face an on-policy issue?**
>
> Interesting point! After a reasoning reward model has been trained, it can be used to train another model's policy. The only limitation in the current implementation is that the tokenisers of both models are consistent. We first test this experiment by swapping the *trained* discriminators with the policy models of the same family (i.e. `Qwen2.5-3B` <--> `Qwen2.5-7B` and `Llama3.2-3B` <--> `Llama3.1-8B`) and try to retrain the policy models from scratch, while keeping the reward models frozen.
>
> We observe that reward discriminators trained in our AIRL setup specialise to the policy they are co-trained with and transfer poorly to new policies. We report this in **Appendix D.8, page 39**.
>
>
> ### Changes to the Manuscript
>
> Based on the reviewers' comments, we have made the following changes to the updated manuscript:
>
> - New baselines on MedReason: Appendix C (pages 23-30)
> - GSM8K training without reward normalisation: Appendix D.6 (page 37)
> - GSM8K training with erroneous expert reasoning traces: Appendix D.7 (page 38)
> - GSM8K interchanging reward model: Appendix D.8 (page 39)
>
> We hope that our response addresses all the issues raised by the reviewer and highlights the novelty, effectiveness, and generalisability of our paper. Thanks to the reviewer's insightful suggestions, we were able to significantly improve the manuscript and gain additional insights into the methodology!

---

### Official Review · Reviewer_HGz5 · 2025-11-01

**Soundness:** 3
**Presentation:** 3
**Contribution:** 3
**Rating:** 4
**Confidence:** 4

**Summary:**

This paper explores an interesting research question and proposes a framework based on adversarial inverse reinforcement learning (IRL), where a discriminator learns to distinguish expert reasoning trajectories from model-generated ones, and its logits are used as dense rewards for each token.
These rewards are then used both for training and inference-time reranking of reasoning candidates. Experiments on GSM8K with Llama3 and Qwen2.5 show that the learned reward correlates with correctness, improves reranking, and offers interpretable error localization.

**Strengths:**

The paper addresses an important and underexplored direction, i.e., learning process-level reasoning rewards.
The method is clearly motivated, well-formulated, and conceptually elegant.
It unifies training, evaluation, and interpretability under a single learned dense reward.
The use of IRL in reasoning is novel, bridging reinforcement learning theory with LLM reasoning supervision.
The evaluation framework is also systematic and covers both performance and interpretability aspects.

**Weaknesses:**

There are also several important concerns that need to be addressed.
While the paper shows performance improvements from using the learned reward for reranking, it does not include comparisons against other SOTA mathematical reasoning baselines. This makes it unclear whether the observed gains truly come from the proposed inverse-RL reward or if they could be achieved by existing reasoning-specific techniques such as verifier-based reranking or process reward models.

The experiments are limited to GSM8K, which restricts the generality of the conclusions. Given that the central claim is about learning a reward that captures general reasoning process quality, evaluating on at least one non-mathematical or multi-domain dataset would make the evidence much more convincing. The lack of such breadth is a major limitation, even though the authors acknowledge this point in their Limitations.

The training setup involves alternating updates between the policy and the discriminator, yet the paper provides little analysis of training stability, variance, or convergence behavior. Important hyperparameters are not ablated, and the robustness of the discriminator to noisy or imperfect demonstrations remains untested.

**Questions:**

Please refer to the concerns in Weaknesses.

---

> ### Author Response · Authors · 2025-11-26
>
> Thank you very much for the positive comments and the constructive feedback! We are glad to see that the reviewer considers the problem underexplored and that our paper offers a novel approach. Additionally, we wish to extend our apologies for the late reply; the follow-up experiments took longer than expected. **We have uploaded a new manuscript, and all our changes are highlighted in blue.**
>
> We address the weaknesses that the reviewer mentioned below:
>
> **W1) Comparison against SOTA Reasoning Baselines**
>
> Thank you for this comment. We do compare our method against Outcome Reward Models (ORM), which use GRPO on verifiable outcomes, and also against knowledge distillation with CoTs using Supervised Finetuning (SFT) in Appendix C.1, Table 2.
>
> In this table, we can see that our method does not outperform ORMs. While these results might look discouraging at first, in our opinion, this is to be expected, as the ORM has the chance to be trained on a verifiable reward, meaning we *have access to the correct solution*. The AIRL approach nevertheless achieves competitive results solely by trying to *infer the reward*. Just as in standard RL, a policy will almost always be better trained if we have access to the ground-truth reward function, rather than in an inverse RL approach, where the reward function must first be learned before the policy can be induced.
>
> **W2) Only validated on GSMK8**
>
> Thank you for this feedback. We agree that validation on GSM8K alone is insufficient to demonstrate the generalisability of our paper.  To this end, we have conducted additional benchmark experiments on the **MedReason Dataset** [1], to prove that our method also works in more difficult high-stakes environments, like medical decision-making.
>
> We ran all experiments on the original model combinations in the MedReason dataset and report the results **Appendix C (pages 23-30)**. While the training is noisier than with a simple dataset like GSM8K, we can see that, e.g., the reranking still works very well, and the large models still show a strong correlation with medical correctness. Moreover, the reranking strategy still provides significant improvements, especially on pass1|N. We find the results particularly interesting in a high-stakes environment, where verifiable rewards are not always guaranteed, and interpretability is crucial.
>
> We hope that these additional results demonstrate further the generalisability of our method.
>
> [1] https://arxiv.org/abs/2504.00993: MedReason: Eliciting Factual Medical Reasoning Steps in LLMs via Knowledge Graphs
>
> **W3) Ablation, Convergence, Imperfect Demonstrations**
>
> Thank you for this comment. Regarding the ablations, we additionally run the following experiments:
>
> - only using an outcome reward signal in AIRL (Appendix D.5, page 36)
> - removing reward normalisation (Appendix D.6, page 37)
> - imperfect/noisy demonstrations (Appendix D.7, page 38)
> - changing the reward model to a new policy model (Appendix D.8, page 39)
>
> Regarding the convergence of the algorithm, our primary contribution is not a new IRL algorithm but the **adaptation** of existing adversarial frameworks (such as AIRL/GAIL) to learn a **dense, token-level reasoning reward** for LLMs. The paper's core is demonstrating the *utility* of this learned reward via our four desiderata (training signal, inference-time assistance, correctness over style, and error localisation.
>
> **Changes made to the Manuscript**
>
> Based on the reviewer’s comment, we have made the following changes to the manuscript:
>
> - Experiments for MedReason (Appendix C, pages 23-30)
> - Ablation only using outcome reward signal (Appendix D.5, page 36)
> - Ablation without normalisation of rewards (Appendix D.6, page 37)
> - Ablation for imperfect demonstrations (Appendix D.7, page 38)
> - Ablation with changing reward model (Appendix D.8, page 39)
>
> We thank the reviewer again for their constructive feedback and hope that our additional experiments further convince them of the generalisability and novelty of our method. Thanks to all the strong ideas from the reviewer, we were able to improve the manuscript a lot and gain extra insights into the methodology.

---

### Official Review · Reviewer_fXdx · 2025-11-02

**Soundness:** 3
**Presentation:** 3
**Contribution:** 3
**Rating:** 6
**Confidence:** 4

**Summary:**

This paper proposes a novel approach to enhancing multi-step reasoning in Large Language Models by learning a dense, token-level reward model via adversarial Inverse Reinforcement Learning. Instead of imitating expert reasoning traces through supervised fine-tuning, the method infers a reward function that evaluates the quality of intermediate reasoning steps. The learned reward serves a dual purpose: as a training signal to optimize a reasoning policy, and as an inference-time critic for reranking sampled traces under a fixed compute budget. The authors demonstrate that this reward correlates with correctness rather than stylistic conformity and provides interpretable, token-level diagnostics. Empirical evaluation is primarily conducted on the GSM8K mathematical reasoning benchmark using Llama and Qwen model families.

**Strengths:**

The work creatively reformulates adversarial IRL for LLM reasoning, moving beyond pure imitation learning (SFT) and outcome-based reinforcement learning (e.g., GRPO). The idea of learning a dense, process-oriented reward from demonstrations is a notable contribution.
The paper is generally well-written, with a clear problem formulation, method description, and experimental setup. Figures are effective in illustrating key points.
The concept of a unified, reusable process-level critic for training, inference, and diagnosis is promising and could influence how reasoning is optimized in LLMs.

**Weaknesses:**

the primary empirical validation is confined to a single domain—mathematical reasoning on GSM8K. While GSM8K is a standard benchmark, the claims of the method's general potential for "multi-step reasoning" would be significantly strengthened by evaluation on a more diverse set of reasoning tasks (e.g., logical deduction, commonsense reasoning, code generation). The current evidence is insufficient to conclude that the approach generalizes beyond arithmetic problems.

**Questions:**

Given that the evaluation is primarily on GSM8K, what evidence or ablation studies can the authors provide to support the claim that the learned reasoning reward captures general reasoning principles rather than features specific to mathematical word problems? Would the method be effective on a fundamentally different reasoning task, such as logical reasoning (e.g., on PrOntoQA or ProofWriter) or multi-hop QA?

---

> ### Author Response · Authors · 2025-11-26
>
> We wish to thank the reviewer for their positive feedback. We are glad to see that our method is seen as novel and that our paper is well written. Additionally, we wish to extend our apologies for replying so late; running the follow-up experiments took longer than expected. **We have uploaded a new manuscript, and all our changes are highlighted in blue.**
>
> We address the reviewer’s concern below:
>
> **W1) Validated only on GSM8K**
>
> Thank you for this feedback. We agree that validation on GSM8K alone is insufficient to demonstrate the generalisability of our paper.  To this end, we have conducted additional benchmark experiments on the **MedReason Dataset [1]** to demonstrate that our method also works in more difficult, high-stakes environments, such as medical decision-making.
>
> We ran all experiments on the original model combinations in the MedReason dataset and report the results in **Appendix Section C (page 23-30, don't worry, mostly images)**. We. We can see that our results are mostly consistent with those reported for GSM8K. While the training is noisier than with a simple dataset like GSM8K, we can see that, e.g., the reranking still works very well, and the large models still show a strong correlation with medical correctness. We find the results particularly interesting in a high-stakes environment, where verifiable rewards are not always guaranteed, and interpretability is crucial.
>
> We hope that these additional results demonstrate further the generalisability of our method.
>
> [1] https://arxiv.org/abs/2504.00993: MedReason: Eliciting Factual Medical Reasoning Steps in LLMs via Knowledge Graphs
>
> We hope that these additional results further demonstrate the paper's generalisability and that we have addressed the reviewer’s concern. Thanks to the reviewer's input, we were able to improve the manuscript significantly!
>
> ### Changes to the Manuscript:
>
> - We have updated the new manuscript with the experiments for MedReason: Appendix C (pages 23-30)

---

> > ### Comment · Reviewer_fXdx · 2025-11-27
> >
> > I thank the authors for their detailed response and for conducting additional experiments on the MedReason dataset to address the concern regarding generalization.
> >
> > While I appreciate the new results presented in Appendix C, I have a specific concern regarding Figure 18. It appears that both the average Reward value and the Correctness score exhibit a downward trend as the training steps increase. This is quite counter-intuitive, as we typically expect the performance to improve or at least stabilize during training.
> >
> > Could the authors provide further analysis or explanation for this phenomenon? Specifically, why does the model performance seem to degrade as training progresses on this dataset? Does this indicate instability in the adversarial training process when applied to more complex or "noisier" domains like medical reasoning, compared to GSM8K?

---

> ### Author Response · Authors · 2025-11-27
>
> Thank you for your comment!
>
> We agree that this is counterintuitive. It is especially noticeable on the Qwen2.5-7B policy. We are not quite sure what the precise cause of this phenomenon is; however, we do have some hypotheses:
>
> - In adversarial IRL, the reward model and policy form a saddle-point optimisation problem, so neither the average reward nor correctness is guaranteed to increase monotonically. Based on the observed losses, we believe our discriminator may be too strong.
> - On MedReason, where we do not perturb expert traces, the discriminator could overfit stylistic cues (formatting, structure) that are more variable than correctness itself.
>
> **We believe the downward trend in Fig. 18 is due to known instabilities in adversarial IRL on complex, noisy domains rather than a fundamental failure of the framework**. However, some potential remedies might include training it longer or introducing a KL penalty on the policy model to help avoid strong model degradation.
>
>
> Thanks again for your input; it is much appreciated.

---

### Official Review · Reviewer_APqg · 2025-11-04

**Soundness:** 3
**Presentation:** 3
**Contribution:** 2
**Rating:** 2
**Confidence:** 4

**Summary:**

This paper proposes reframing large language model reasoning as an inverse reinforcement learning problem, where a dense, token-level reward model is learned from expert demonstrations to provide process supervision. The reward model serves dual purposes: as a training signal for optimizing a reasoning policy and as a critic for reranking sampled traces at inference time. Contributions include demonstrating that this approach prioritizes correctness over stylistic imitation, improves predictive performance via reranking, and enables interpretable error localization, evaluated on GSM8K with Llama3 and Qwen2.5 models.

**Strengths:**

1. The paper creatively adapts adversarial IRL to LLM reasoning, shifting from traditional supervised fine-tuning to learning dense rewards from demonstrations, which could inspire new ways to handle multi-step tasks without explicit reward engineering.

2. The presentation is strong, with well-illustrated figures (e.g., Figure 1 overview) and tables (e.g., Table 1 comparison) that effectively communicate the approach and desiderata.

3. Ignoring implementation challenges, using IRL to construct dense rewards offers a promising idea for unifying training and inference in reasoning tasks, potentially reusable across domains like math and logic.

**Weaknesses:**

1. Limited Novelty: The core approach appears as a straightforward application of AIRL (Fu et al., 2018), with foundational elements like GAN-style training or PRM + GRPO being well-established and widely used; it lacks significant extensions to differentiate from prior IRL adaptations in LLMs.

2. Insufficient Experiments: Evaluations are confined to GSM8K, a dataset that has lost much of its challenge for modern models, raising doubts about generalization to harder benchmarks like MATH or code generation; additional datasets would strengthen claims.
Training Instability: While IRL is conceptually appealing, joint training of discriminator and policy often struggles with convergence due to shifting distributions, exacerbating learning difficulty—as the authors acknowledge in limitations; more robust stabilization techniques could help.

3. Readiness for Publication: The paper contains several typos and inconsistencies (e.g., line 98: "y1" should be "y2"; line 340: "Amswer" should be "Answer"), suggesting it needs polishing before submission.

4. Weak Inference Comparisons: For test-time scaling, comparisons are limited to random ranking; lacking baselines like outcome reward models (ORMs), process reward models (PRMs), or heuristic methods (e.g., selecting shortest responses) makes it hard to assess the method's superiority.

**Questions:**

For more complex scenarios, such as problems that a reasoning model might solve through reflection, where intermediate errors could be acceptable if the final answer is correct, is this dense reward still necessary?

---

> ### Author Response · Authors · 2025-11-26
>
> We thank the reviewer for their constructive feedback. We’re glad the reviewer finds our methodology creative and a promising idea, and that our presentation is strong and well illustrated. Additionally, we wish to extend our apologies for replying so late; running the follow-up experiments took longer than expected. **We have uploaded a new manuscript and all our changes are highlighted in blue**
>
> We address the concerns and questions below:
>
> **W1) Limited Novelty**
>
> Thank you for raising this concern. While we acknowledge the reviewer’s point of view, we must respectfully disagree. To the best of our knowledge, this is one of the first approaches to apply inverse reinforcement learning to large language model reasoning. Given the success and potential parallels to RLHF, we decided to go with Adversarial Inverse Reinforcement Learning by Fu et al. (2019). The core contribution of this paper is the extraction of a dense reasoning reward model directly learned from reasoning traces, and its subsequent use for inference improvement and Chain-of-Thought diagnosis. Moreover, while the core algorithm originates from Fu et al. 2019, we had to expand it to incorporate LLM-specific considerations, such as dense reward and advantage calculations, and to augment the expert data with errors.
>
> **W2) Insufficient Experiments**
>
> Thank you for this feedback. We agree that validation on GSM8K alone is insufficient to demonstrate the generalisability of our paper.  To this end, we have conducted additional benchmark experiments on the **MedReason Dataset** [1] to prove that our method also performs well in more difficult, high-stakes environments, such as medical decision-making.
>
> We ran all experiments on the original model combinations in the MedReason dataset and report the results in **Appendix Section C (page 23-30)**. We can see that our results are mostly consistent with those reported for GSM8K. The reward and correctness is (as expected) noisier during the optimisation than with a simple dataset. However, the re-ranking still performs very strongly. We find the results particularly interesting in a high-stakes environment, where verifiable rewards are not always guaranteed, and interpretability is crucial.
>
> We hope that these additional results demonstrate further the generalisability of our method.
>
> [1] https://arxiv.org/abs/2504.00993: MedReason: Eliciting Factual Medical Reasoning Steps in LLMs via Knowledge Graphs
>
> **W3) Readiness for Publication**
>
> Thank you for pointing out the typos in Figure 1 and Figure 3a! Our grammar correction software did not pick up on these typos because it unfortunately does not parse the content of figures. We have corrected these mistakes in the newest version of the manuscript.

---

> > ### Author Response · Authors · 2025-11-26
> >
> > **W4) Weak Inference Comparisons**
> >
> > Thank you for pointing this out. We agree that heuristics, such as length, could be a way to rerank sampled responses. We run an experiment evaluating the dense reward model against length, as proposed by the reviewer, and find that it still outperforms the heuristic. We report the experiments in Appendix C.9 of the new manuscript.
> >
> > We can see that, for length-re-ranked heuristics (longer sequences are preferred), this heuristic performs worse on the Llama3.2-3B and Llama3.1-8B policy models. For Qwen2.5-3B, there appears to be no statistically significant performance improvement over random reranking. For Qwen2.5-7B, heuristic reranking performs worse than the dense reward and random reranking, especially on pass@1|16.
> >
> > Regarding ORMs or PRMs, we believe this would not be a fair comparison with our learned reward, because both approaches require a verification rule to confirm that the answer (or the intermediate step for PRMs) is correct. If one has access to such a verifier, one could sample until the first correct answer is obtained. However, with the learned reward model, as in RLHF, there is no way to determine whether any of the sampled responses are accurate. The learned reward indicates which sampled response is most likely correct.
> >
> > **Q1) Complex Scenarios**
> >
> > > For more complex scenarios, such as problems that a reasoning model might solve through reflection, where intermediate errors could be acceptable if the final answer is correct, is this dense reward still necessary?
> > >
> >
> > This is an interesting question, and we believe it potentially goes even beyond the dense reasoning reward. As in, do we need Reasoning and Chain-of-Tokens at all if we only care about the correctness of the response?
> >
> > Well, if we solely care about the response correctness and nothing else, then the dense reasoning reward is not necessary. However, we wish to underline again that the benefit of having a dense reasoning reward model is to diagnose and inspect reasoning traces, and to quantify when a Chain-of-Thought potentially goes wrong. With the dense reasoning reward, we have gained a better understanding of which parts receive high rewards and which receive low rewards.
> >
> > ### Actions and Changes to the Manuscript:
> >
> > - Corrected typos in Figures
> > - Included results for experiments on MedReason: Appendix C (pages 23-30)
> > - Included Length-Heuristic Experiment: Appendix D.9 (page 41)
> >
> > We wish to thank the reviewer for their constructive insights, which have significantly improved our manuscript.

---

### Author Response · Authors · 2025-11-30
**Global Response: Changes and Discussion**

We are grateful to the reviewers for their insightful feedback. There is a broad consensus that the problem is important and that our approach is novel.

- APqg: *“[...], using IRL to construct dense rewards offers a promising idea for unifying training and inference in reasoning tasks”*
- fXdx: *“The idea of learning a dense, process-oriented reward from demonstrations is a notable contribution.”*
- HGz5: *“The paper addresses an important and underexplored direction [...]. The method is clearly motivated, well-formulated, and conceptually elegant.”*
- RjAM: *“The token-level dense reward model learned [...] paper breaks through the limitation of separating training signals and inference auxiliary tools in traditional methods.”*
---
### Changes and Discussion after the Rebuttal

In the uploaded PDF, we have highlighted all changes in the main text in **blue**. We implemented a set of focused revisions that address all reviewer concerns, including additional benchmarks, ablations, and clarifications of our adversarial IRL setup.

**Major change summary.**
1. Demonstrate that our adversarial inverse reinforcement learning framework extends beyond GSM8K by adding a full set of experiments on the **MedReason** medical reasoning benchmark (Appendix C), showing that the learned dense reward remains useful for inference-time assistance and interpretability in a high-stakes domain.
2. Provide a richer ablation suite on GSM8K isolating the effect of using only an outcome reward, removing reward normalisation, injecting imperfect expert demonstrations, and transferring the learned reward to different policy models (Appendix D.5–D.8).
3. Clarify our positioning with respect to ORMs/PRMs and standard reasoning baselines, explicitly discuss the non-stationarity and limitations of adversarial IRL in our setting, and fix typos and polish figures.

We believe that these changes further strengthen the empirical evidence and provide a clearer picture of when and how the learned reasoning reward is most effective.

---

### Summary of changes and rebuttal discussion

| **Change** | **Purpose** | **Reviewer(s) addressed** | **Summary** |
| --- | --- | --- | --- |
| MedReason benchmark experiments (Appendix C) | Demonstrate generalisability beyond GSM8K and behaviour in high-stakes domains | APqg, fXdx, HGz5, RjAM | Added a full set of MedReason experiments for all backbone pairs, including training dynamics, reranking performance, correlation analyses, and qualitative dense-reward heatmaps. |
| Discussion of AIRL non-stationarity and limitations | Clarify source of reward fluctuations and scope of our contribution | RjAM | Made explicit that our goal is to adapt existing adversarial IRL (AIRL/GAIL) to dense token-level rewards, and added discussion that non-stationarity is an inherent trade-off of the adversarial setup. |
| Outcome-only AIRL ablation (Appendix D.5) | Separate the effect of dense reasoning rewards from outcome-only training | HGz5, RjAM | Trained AIRL using only a terminal outcome reward, showing slightly stronger performance but losing token-level interpretability, highlighting the trade-off between accuracy and dense diagnostic signal. |
| No-normalisation ablation (Appendix D.6) | Study stability of the training signal | HGz5, RjAM | Removed reward normalisation during training and observed substantial instability and degradation, supporting our design choice to normalise rewards. |
| Imperfect/noisy expert demonstrations (Appendix D.7) | Test robustness to erroneous expert traces | HGz5, RjAM | Corrupted a fraction of expert CoTs on GSM8K and quantified the often resulting performance drop, illustrating sensitivity to expert quality and documenting this limitation. |
| Reward transfer across policies (Appendix D.8) | Address potential on-/off-policy issues | HGz5, RjAM | Swapped discriminators between models of the same family (e.g., Qwen2.5-3B ↔ Qwen2.5-7B, Llama3.2-3B ↔ Llama3.1-8B) and found that reward models specialise to the co-trained policy and transfer poorly, clarifying when reuse is feasible. |
| Length-based reranking heuristic (Appendix D.9) | Compare against simple inference-time baselines | APqg | Implemented reranking by response length and showed that our dense reward outperforms length-based selection and, in some cases, even random reranking, especially on pass@1|16 samples. |
| Clarified comparison to ORMs/PRMs and reasoning baselines | Position our method relative to SOTA reasoning approaches | APqg, HGz5 | Explained why ORMs/PRMs, which rely on verifiable outcome checks, operate in a different setting than our learned reward; highlighted competitive performance given that our reward must be inferred from demonstrations. |
---

### Thank you

We thank all reviewers for their thoughtful and constructive feedback. Their comments led to substantial improvements in our experiments, analysis, and presentation, and helped us better understand the strengths and limitations of our approach.

---

### Meta-Review · Area_Chair_rBLS · 2026-01-05

**Summary:**

The paper proposes to learn a dense a token-level reasoning reward model for LLM via adversarial IRL from expert demonstrations. Reviewers primary focused on the lack of study on training stability (and expect it to have instability), limited evaluations beyond GSM8K and a very controlled evaluation settings (only math etc). Reviewers score were mixed: 2 rejects, marginally below threshold, one marginally above threshold.

**Reviewer Concerns:**

Rebuttal strengthened the empirical evaluation section with additional benchmarks and ablations. However, concerns on novelty, stability or generality were not fully resolved.

**Reviewer Scores:**

Mostly will maintain the score.
APqg: 2 -> 2
fXdx: 6 -> 6
HGz5: 4 -> 4
RjAM: 2 -> 2

---

### Decision · Program_Chairs · 2026-01-26

Reject